# Protective Effects of Phosphatidylcholine against Hepatic and Renal Cell Injury from Advanced Glycation End Products

**DOI:** 10.3390/medicina58111519

**Published:** 2022-10-25

**Authors:** Jihye Choi, Inbong Song, Sangmin Lee, Myungjo You, Jungkee Kwon

**Affiliations:** 1Department of Laboratory Animal Medicine, College of Veterinary Medicine, Jeonbuk National University, Iksan-si 54596, Jeollabuk-do, Korea; 2Laboratory of Veterinary Parasitology, College of Veterinary Medicine, Jeonbuk National University, Iksan-si 54596, Jeollabuk-do, Korea

**Keywords:** diabetes, hepatic and renal injury, phosphatidylcholine

## Abstract

*Background and Objectives:* Receptors of the advanced glycation products (RAGE) are activated to promote cell death and contributes to chronic diseases such as diabetes and inflammation. Advanced glycation end products (AGEs), which interact with RAGE are complex compounds synthesized during diabetes development and are presumed to play a significant role in pathogenesis of diabetes. Phosphatidylcholine (PC), a polyunsaturated fatty acid found in egg yolk, mustard, and soybean, is thought to exert anti-inflammatory activity. We investigated the effects of PC on AGEs-induced hepatic and renal cell injury. *Materials and Methods:* In this study, we evaluated cytokine and NF-κB/MAPK signal pathway activity in AGEs induced human liver (HepG2) cells and human kidney (HK2) cells with and without PC treatment. *Results:* PC reduced RAGE expression and attenuated levels of inflammatory cytokines and NF-kB/MAPK signaling. Moreover, cells treated with PC exhibited a significant reduction in cytotoxicity, oxidative stress, and inflammatory factor levels. *Conclusions:* These findings suggest that PC could be an effective functional material for hepatic and renal injury involving with oxidative stress caused by AGEs during diabetic conditions.

## 1. Introduction

Advanced glycation end products (AGEs) are organic molecules with various structural and functional characteristics that are involved in the development and progression of chronic degenerative diseases such as diabetes [1]. When AGEs are produced in the body, they interfere with insulin-mediated metabolic reactions and are involved in inflammation-mediated cytokines production through promotion of oxidative stress [2]. AGEs are related to the production of reactive oxygen species (ROS), which are mediators of vascular dysfunction [3]. AGEs bind to receptors of AGEs (RAGE) on the cell surface to regulate various cell processes [4]. RAGE is a multiple ligand-receptor expressed in various tissues, including nerve tissue in the vascular system and heart, and various cells including macrophages and microglia [1].

Several studies have shown that the interaction between AGEs and RAGE induces oxidative stress generation and vascular inflammation, platelet activation, and thrombosis, which play a significant role in the development and progression of vascular complications in diabetes [5,6,7]. RAGE activation induces p38 mitogen-activated protein kinase (MAPK) phosphorylation and activation of nuclear factor kappa B (NF-κB) signaling to cause inflammatory reactions in the body, which increases the levels of inflammatory cytokines such as tumor necrosis factor (TNF)-α, interleukin (IL)-1β, and IL-6, resulting in substantial inflammatory damage in the body [8]. Therefore, inhibition of the interaction between AGEs and RAGE downstream signaling have emerged as treatment strategies for diabetes and diabetes complications [6].

Phosphatidylcholine (PC), a polyunsaturated fatty acid, is a component of biological membranes and can be obtained in the diet from various resources such as egg yolk, mustard, sunflower, and soybean [9]. PC has been found to have anti-inflammatory [10], antioxidant [11,12], and anti-Alzheimer’s [13] effects. The role of PC on oxidative stress has been reported in cell and animal experiments [11,12], but the role of PC in kidney and liver cells based on AGEs damage have not been reported.

RAGE stimulation induces activation of NF-κB, a major transcription factor regulating the inflammatory response, which causes diabetes target cell stress and chronic dysfunction [14]. RAGE activation promotes the secretion of cytokines through toll-like receptor (TLR) and NF-κB signaling pathways and indicates an immune state vulnerable to chronic inflammation resulting in diseases such as type 2 diabetes [15]. RAGE is also involved in various intracellular signaling pathways, especially ERK1/2 and p38 are considered to be AGEs-sensitive among the MAPK pathway [16,17]. RAGE activation by AGEs leads to the production of various inflammatory mediators [18], i.e., chronic inflammation, which results in an impairment of cellular function.

In this study, the protective effects of PC on RAGE/MAPK/NF-κB pathway activation in AGEs-induced cell conditions were investigated using HepG2 human liver and HK2 human kidney cells. Additionally, HepG2 and HK2 cells were evaluated for ROS production in AGEs-induced oxidative stress, and inflammatory markers of the MAPK/NF-κB signal transmission pathway were assessed. In this study, PC was evaluated as a potential candidate for diabetes treatment to suppress inflammatory reactions in liver and kidney cells and to respond to diabetic diseases.

## 2. Materials and Methods

### 2.1. Chemicals

Phosphatidylcholine (PC), fetal bovine serum (FBS), penicillin and streptomycin antibiotics (P/S), and MTT assay kits were purchased from Sigma-Aldrich (St. Louis, MO, USA). Phosphate buffer saline (PBS) and AGEs (modified with BSA) were purchased from BioVision (2221-10, Milpitas, CA, USA). The 2′,7′-Dichlorofluorescin diacetate (DCFDA) stain was purchased from Invitrogen (Middlesex County, MA, USA). Dulbecco’s Modified Eagle Medium (DMEM) was purchased from HyClone (Logan, UT, USA). RAGE antibody was purchased from Santa Cruz Biotechnology (SC-365154, Santa Cruz, CA, USA). NK-κB, p- NK-κB, and Histon H3 antibodies were purchased from Abcam (Cambridge, MA, USA). ERK, p-ERK, JNK, p-JNK, COX-2, iNOS, and β-actin antibodies were purchased from Cell Signaling (Danvers, MA, USA). Goat anti-rabbit IgG and goat anti-mouse IgG secondary antibodies were purchased from Millipore (Billerica, MA, USA).

### 2.2. Cell Culture

HepG2 human liver cells (HB-8065) and HK2 human kidney cells (CRL-2190) obtained from the American Type Culture Collection (ATCC, Manassas, VA, USA). The cells were cultured in DMEM medium containing 10% fetal bovine serum and 1% penicillin and streptomycin. The HepG2 and HK2 cells were cultured at 37 °C under 5% CO_2_ in a humidified incubator.

### 2.3. Cell Viability Assays

The MTT assay kit (Sigma-Aldrich, St. Louis, MO, USA) was employed to determine cell viability according to manufacturer instructions. HepG2 and HK2 cells (1 × 10^4^ cells/well) were plated in 96-well plates (Thermo Sciences, Waltham, MA, USA). The cells were treated with either PC (5–100 μg/mL) or AGEs (100 μg/mL) for 24 h or were pre-treated for 1 h to cells with PC (5–100 μg/mL) and then with 100 μg/mL of AGEs for 24 h. A 10 μL (500 μL) aliquot of MTT solution was added to each well after the specified incubation time, and the cells were further incubated for 2 h. Cell viability was determined by absorbance at 540 nm using a BioTek microplate reader (Winooski, VT, USA).

### 2.4. Intracellular ROS Determination

The amount of ROS produced in HepG2 and HK2 cells was measured using fluorescence analysis. The DCFDA/H2DCFDA-cellular ROS assay kit (ab113851, Abcam, Cambridge, UK) was used to measure the effect of ROS production on PC-treated cells. HepG2 and HK2 cells at 4 × 10^5^ cells/well were grown in 48-well plates treated with PC. After confirming cell adhesion, PC was added at two concentrations (10, 100 μg/mL), followed 4 h later by the addition of AGEs (100 μg/mL) to induce oxidative stress. After a 24 h incubation, the cells were washed with PBS and stained with DCFDA (20 μM) for 20 min at 37 °C in the dark. Intracellular ROS was measured using fluorescence at a wavelength of 530 nm.

### 2.5. Western Blotting

HepG2 and HK2 cells were treated with BSA (control) or PC (10, 100 μg/mL), followed 4 h later by, addition of 100 μg/mL AGEs to induce inflammatory conditions- and incubated for 24 h. After growth, the crude cells extracts were prepared using lysis buffer, and 30 μg of protein was separated using sodium dodecyl sulfate-polyacrylamide gel electrophoresis (SDS-PAGE) with an 8–15% gel. The isolated proteins were transferred to a polyvinylidene fluoride (PVDF) membrane. The membrane was blocked with 5% skim milk for 1 h and, then incubated with the primary antibody (1:1000 *v*/*v* ratio) in 1% skim milk in PBS-T at 4 °C overnight. The primary antibodies used were RAGE (Santa Cruz, CA, USA), p-NF-κB, NF-κB, Histone H3 (Abcam, Cambridge, MA, USA), p-ERK, ERK, p-p38, p38, p-JNK, JNK, COX-2, iNOS, and β-actin (Cell signaling, Danvers, MA, USA). Finally, membranes were incubated with goat-anti-rabbit IgG or goat-anti-rabbit (1:5000 *v*/*v* ratio) as the secondary antibody. Each antigen–antibody complex was visualized using Super Signal West Dura Extended Duration Substrate on a Chemi Imager system (Alpha Innotech, San Leandro, CA, USA). The β-actin antibody as a control was used to ensure equal sample loading and protein transfer.

### 2.6. Statistical Analysis

All data are expressed as mean ± SEM, and all experiments were repeated more than three times to foster data reliability. Statistical analysis was analyzed using the GraphPad Prism 5 program (GraphPad Software version 5, San Diego, CA, USA). Differences between the control and experimental groups were analyzed via Student’s *t*-test.

## 3. Results

### 3.1. PC Attenuates ROS Production in AGEs-Induced in HepG2 and HK2 Cells

The effects of PC on ROS production in AGEs-treated HepG2 and HK2 cells were assessed using MTT analysis. AGEs concentrations of 5, 10, and 100 μg/mL were used (Figure 1a,b), and both HepG2 and HK2 cells confirmed a significant decrease in cell viability at 100 μg/mL, the concentration used to assess the effect of PC on cells. The degree of cytotoxicity induced by AGEs was evaluated in HepG2 and HK2 cells, which were then treated with PC to observe the protective effect against cytotoxicity (Figure 1c,d). Although not significant, we observed a relationship between the degree of recovery due to PC concentration.

The protective effect of PC on oxidative stress induced by AGEs was measured at the level of intracellular ROS (Figure 1e,f). HepG2 and HK2 cells treated with AGEs showed an approximately 2.5-fold higher cellular ROS production compared with untreated control cells. ROS production after AGEs treatment was significantly decreased in cells pretreated with PC than in those treated with AGEs alone.

### 3.2. PC Regulates RAGE Expression in AGEs-Treated Cells

The production of RAGE was assessed by Western blot using HepG2 and HK2 cells treated with AGEs at two concentrations (10 and 100 μg/mL) for 24 h. RAGE production in HepG2 and HK2 cells were confirmed at 100 μg/mL, which was set as the concentration for further experimentation (Figure 2a,b). RAGE production in HepG2 and HK2 cells showed a significant concentration reduction when treated with PC (Figure 2c,d).

### 3.3. PC Downregulates Inflammatory Factors and NF-κB in AGEs-Treated Cells

Inflammatory factor levels were analyzed in HepG2 and HK2 cells to evaluate the inhibitory effect of PC in AGE-induced inflammatory conditions. The concentration of cyclooxygenase (COX)-2 was increased in both HepG2 and HK2 cells by AGEs, which showed a significant reduction at both PC concentrations of 10 and 100 μg/mL, confirming results similar to those of cells that were not treated with AGEs (Figure 3a,b). Since the level of inducible nitric oxide synthase (iNOS) affects the formation of basic nitrogen oxide as an inflammatory indicator, the reduction of iNOS production was confirmed in cells treated with PC compared with AGEs alone (Figure 3c). In HK2 cells, the expression of reduced concentration-dependent iNOS was determined that the effective anti-inflammatory action is performed in AGEs-induced inflammatory conditions (Figure 3d). PC treatment also reduced the activity of NF-kB, phosphorylation NF-kB in HepG2 and HK2(Figure 3e,f).

### 3.4. PC Downregulates p38MAPK Signaling in AGEs Treated Cells

AGEs treatment causes inflammatory conditions by inducing activation of the MAPK pathway, which affects the p38 MAPK pathway. The expression of ERK was significantly reduced in both cell lines after treatment with 100 μg/mL pf PC (Figure 4a,b). The expression of p38, which shows the strongest response to AGEs treatment, was reduced in PC-treated HepG2 cells (Figure 4c,d) and HK2 cells (Figure 4c,d). Based on these results, it is judged that PCs affect the inhibition of MAPK pathway activation induced by AGEs and help maintain normal immunity in the body.

## 4. Discussion

In this study, we evaluated the protective role of PCs and its anti-inflammatory properties against AGEs-induced human liver cells (HepG2) and kidney cells (HK2) that are representative metabolic organs. The liver is mainly responsible for physiological absorption, preservation, and detoxification of drugs and other toxins containing protein, carbohydrates, and fat. Conversely, the kidney regulates the excretion and reabsorption of inorganic electrolytes and water to control volume, mineral components, and acidity. In previous studies, the HepG2 and HK2 cell lines were used as model systems for research on cell proliferation inhibition and cell death [19,20,21], and they have also been used to determine the antioxidant efficacy of many compounds [22].

In diabetic conditions, accumulation of AGEs is an important pathophysiological mechanism that induces complications and adverse effects on the body through interaction with RAGE [23]. As a member of the cell surface pattern-recognition receptors (PRRs), RAGE activation results in inflammation in the body [24]. RAGE plays an essential role in acute liver injury, and blocking RAGE signaling alleviate ischemia, toxicity, and bile liver injury [25,26], including kidney disease [27]. We confirmed an increase in RAGE production in this study by treatment with AGEs in HepG2 and HK2 cells, as RAGE recognizes AGEs from cells and mediates the signal response to the body. RAGE-AGE interactions are also influenced by blood glucose concentration and degree of oxidative stress. As the condition persists, production of ROS increases, activating inflammatory factors such as the MAPK pathway and NF-κB [28]. Accumulated evidence has confirmed that AGE binding to RAGE induces a cascade of pathophysiological conditions related to downstream activation of NF-κB, resulting in inflammation [29]. The reduced expression of activated RAGE based on PC treatment is presumed to be due to the inhibitory inflammation effects of PC.

NF-κB plays a crucial role in AGEs-induced inflammatory responses in diabetic conditions. NF-κB regulates the expression of cytokines and mediators-encoding genes, such as interleukin (IL)-1β [30] and tumor necrosis factor (TNF)-α. Generally, NF-κB is located in the cytosol in an inactive form. ROS promotes rapid translocation of active NF-κB into the nucleus, inducing the production of inflammatory enzymes such as iNOS and COX-2 [31]. In addition, NF-κB is a crucial transcription factor for the induction of IL-1β-mediated iNOS gene expression and extracellular signal-related protein kinases, including p38, ERK, and JNK kinase pathway, which induce COX-2 [32].

The results of this study are consistent with previous studies showing that AGEs-induced hepatic and renal injuries that accompany changes in inflammatory signaling. As clearly shown in the PC-treated group, PC significantly decreased the production of p-NF-kB, COX-2, and iNOS (Figure 3a–f). These results indicate that PC significantly decreases AGEs-induced NF-κB inflammation signaling in HepG2 and HK2 cells. Consistently, PC significantly reduced the expression of p-p38, p-ERK, and p-JNK (Figure 4a–f), which suggests that PC significantly decreased AGEs-induced MAPK signaling in HepG2 and HK2 cells. Specifically, PC significantly reduced the production of RAGE induced by AGEs in a dose-dependent manner (Figure 2a–d). Inhibition of RAGE production by PC may be correlated with the increase of cell viability in damaged hepatic and renal cells by accumulating AGEs.

This study confirmed the protective effect of PC treatment in AGEs-induced RAGE kidney cells and hepatocytes. With this confirmation, it is necessary to establish the value of PC as a preventative material for diabetes through mechanistic research.

## 5. Conclusions

This study shows that treatment with PC significantly attenuates the production of inflammatory signaling factors in AGEs-treated hepatic and renal cells. RAGE induction by AGEs was also attenuated by PC in both cell lines. These findings suggest that PC exerts protective effects against hepatic and renal injuries by AGEs, indicating that PC is a potentially helpful candidate for improving diabetic conditions and liver and kidney-associated diseases.

## Figures and Tables

**Figure 1 medicina-58-01519-f001:**
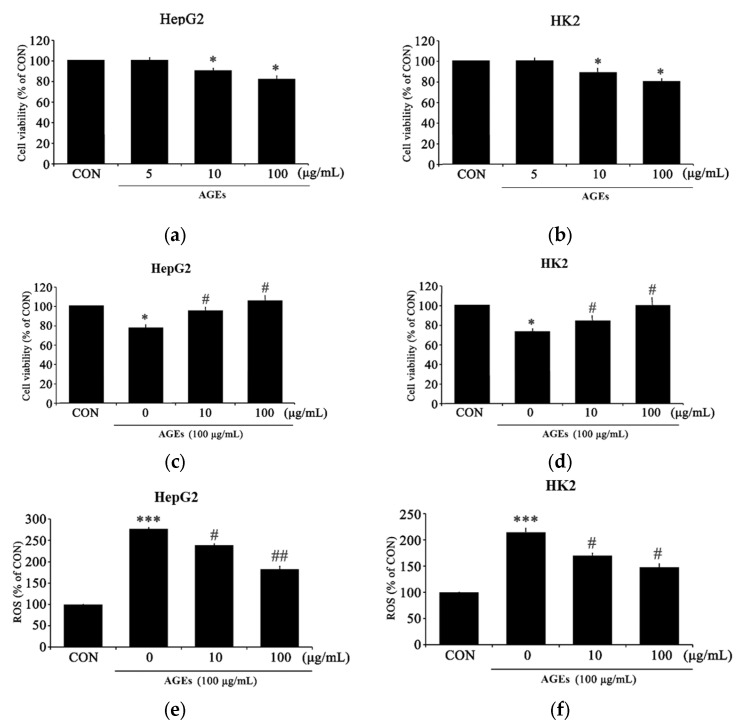
The effects of PC on cell viability and intracellular ROS production in AGEs-treated HepG2 and HK cells. (**a**) Viability of HepG2 cells after treatment with AGEs, (**b**) viability of HK2 cells after treatment with AGEs, (**c**) viability of HepG2 cells pretreated with PC (10 and 100 μg/mL) for 1 h, followed by treatment with AGEs (100 μg/mL) for 24 h, (**d**) viability of HK2 cells pretreated with PC (10 and 100 μg/mL) for 1 h, followed by AGEs (100 μg/mL) for 24 h, (**e**) ROS levels in HepG2 cells pretreated with PC followed by AGE treatment, and (**f**) ROS levels in HK2 cells pretreated with PC followed by AGE treatment. * *p* < 0.05, *** *p* < 0.005 compared with the control (CON); # *p* < 0.05, ## *p* < 0.01 compared with AGE treatment only.

**Figure 2 medicina-58-01519-f002:**
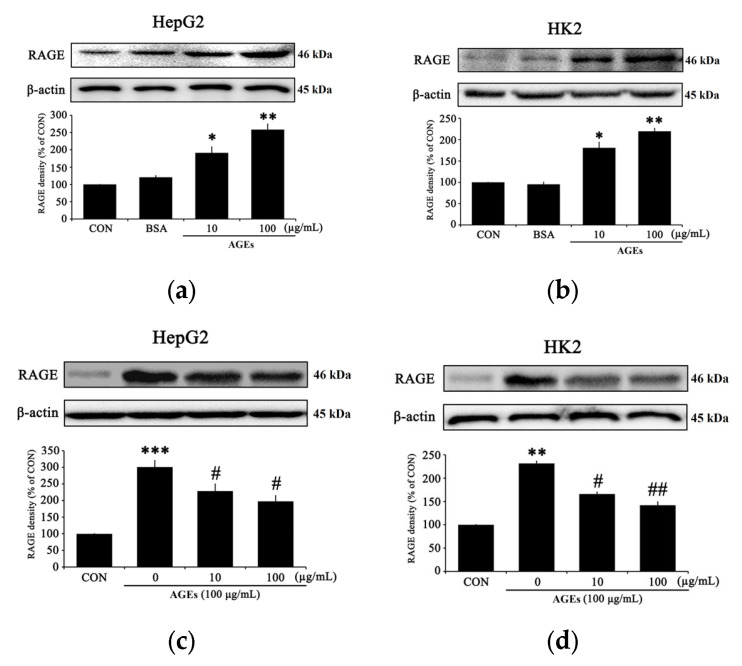
The effect of PC on activation of RAGE in AGEs-treated HepG2 and HK2 cells. (**a**) RAGE protein production in HepG2 cells treated with AGEs, (**b**) RAGE protein production in HK2 cells treated with AGEs, (**c**) inhibition of RAGE activation derived from HepG2 cells of AGEs by PC pretreatment, and (**d**) inhibition of RAGE activation derived from HK2 cells of AGEs by PC pretreatment. * *p* < 0.05, ** *p* < 0.01, and *** *p* < 0.005 compared with the control (CON); # *p* < 0.05, ## *p* < 0.01 compared with AGE-treated cells only.

**Figure 3 medicina-58-01519-f003:**
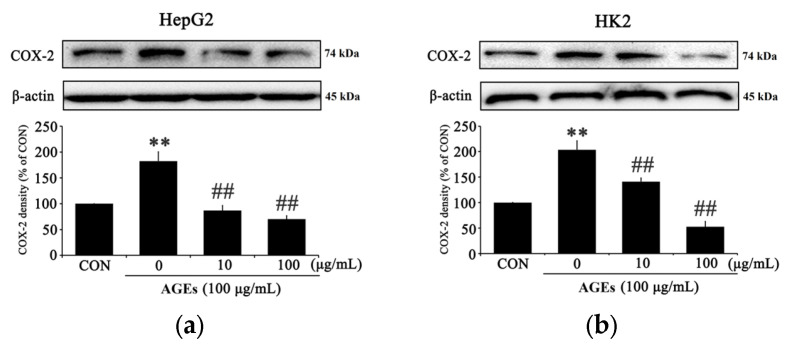
The effect of PC on activation of inflammation-related protein expression in AGEs-treated HepG2 and HK2 cells. HepG2 and HK2 cells pretreated with PC (10 and 100 μg/mL) for 1 h and then with 100 μg/mL AGEs for 24 h. (**a**) COX-2 production in HepG2 cells, (**b**) COX-2 production in HK2 cells, (**c**) iNOS production in HepG2 cells, (**d**) iNOS production in HK2 cells, (**e**) p-NF-κB production in HepG2 cells, and (**f**) p-NF-κB production in HK2 cells. ** *p* < 0.01 compared with the control (CON); # *p* < 0.05, ## *p* < 0.01 compared with AGEs-treated cells only.

**Figure 4 medicina-58-01519-f004:**
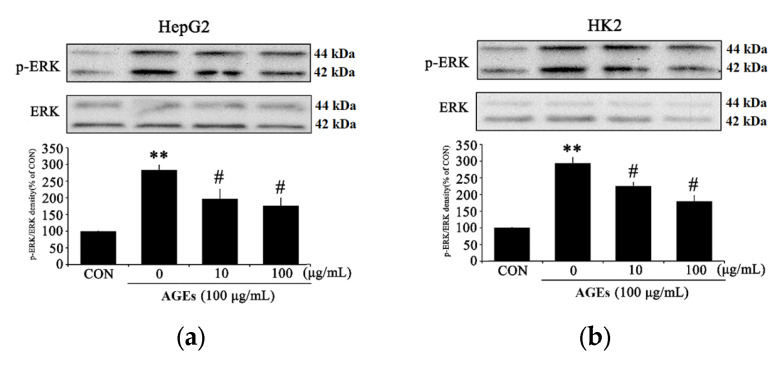
The effect of PC on activation of MAPK signaling in AGEs-treated in HepG2 and HK2 cells. HepG2 and HK2 cells pretreated with PC (10 and 100 μg/mL) for 1 h and then with 100 μg/mL AGEs for 24 h. (**a**) p-ERK/ERK production in HepG2 cells, (**b**) p-ERK/ERK production in HK2 cells, (**c**) p-p38/p38 production in HepG2 cells, (**d**) p-p38/p-38 production in HK2 cells, (**e**) p-JNK/JNK production in HepG2 cells, and (**f**) p-JNK/JNK production in HK2 cells. * *p* < 0.05, ** *p* < 0.01 compared with the control (CON); # *p* < 0.05, ## *p* < 0.01 compared with AGEs-treated cells only.

## Data Availability

Not applicable.

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
