# Peer review of "Protective Effects of Phosphatidylcholine against Hepatic and Renal Cell Injury from Advanced Glycation End Products"

_medicina, 2022, doi:10.3390/medicina58111519_

Round 1

Reviewer 1 Report

The manuscript needs to be edited by a native English Speaker. In its current form, the science cannot be evaluated.

Author Response

Dear Reviewer,

On behalf of the authors, I would like to thank you for providing us the opportunity to improve our manuscript once again. We appreciated the careful reading of our manuscript as well as commenting and suggesting for better manuscript. 

We have carefully rewritten and reorganized our manuscript according to the comments from you. We have asked a native speaker to review our revised manuscript thoroughly again.

We hope that you agree with our manuscript that has been not only throughly revised but also strengthened by your comments. 

Thank you for your kind consideration.

Reviewer 2 Report

The quality of this manuscript is limited by English grammatical inconsistencies, which can be easily remedied by an editor.

Phosphatidylcholine can be metabolized to choline, which has actions other than membrane stabilization, i.e., participation in one-carbon metabolism and agonist activity at alpha7-nicotinic receptors. The agonist activity of choline at alpha-7nicotininc receptors is particularly relevant, because it an anti-inflammatory effect. There are several possible remedies for the authors’ failure to consider these mechanisms. One strategy would be to measure the levels of phosphatidylcholine, choline, and betaine in their system. Measurable levels of betaine would suggest that the one-carbon pathway has been activated. A second strategy would be to use pharmacological blockade, e.g., methyllycaconitine, to assess whether activation of nicotinic receptors is a potential mechanism.

Author Response

Dear Reviewer,

On behalf of the authors, I would like to thank you for providing us the opportunity to improve our manuscript once again. We appreciate the careful reading of our manuscript as well as commenting and suggesting for a better manuscript

We have carefully rewritten and reorganized our manuscript according to the comments from you. As a matter of fact, we have asked a native speaker to review our revised manuscript thoroughly again.

Also, we will take the points you pointed out and conduct further research. On behalf of the researchers, thank you for helping us develop our research.

We hope that you agree with our manuscript that has been not only thoroughly revised but also strengthened by your comments.

Thank you for your kind consideration.

Round 2

Reviewer 1 Report

The manuscript has improved and can now be reviewed appropriately.

Line 11: The sentence ” RAGE is activated to promote cell death and contributes to…tumors” needs to be modified. In tumors, activation of RAGE results in tumor cell survival, not cell death.

Line 16: The statement “we investigated the effects of PC on acute renal failure and hepatic dysfunction” is not correct if it refers to the present study. In the current manuscript, the authors only tested the effects of PC on cells. The statement needs to be modified.

Line 29-31: The statement “AGEs…are involved in generation infection….” is not correct, please modify.

Line 49: Please add “the role of” before PC to complete the sentence.

Line 58: Modify the sentence that ends with….”are known to AGEs.”In its current form, the sentence is incomplete and doesn't make sense.

Line 72: Please confirm that the molecule used to activate RAGE is AGE conjugated with bovine serum albumin, do the authors intended to say that the AGEs were generated from bovine serum albumin ? what was the catalog number of this product from BioVision? Please indicate it in the Materials and Methods section. In this same section, please also indicate the catalog number of the anti-RAGE antibody used in Western blots.

In general, in all images showing the bands from a WB, please indicate the molecular weight of the band that is shown. Also indicate if this MW is the anticipated MW.

In the legend of Figure 2, c) and d), please replace “inhibition of RAGE activation” by inhibition of RAGE expression.

Line 180: Please modify the sentence starting with “AGEs-treated causes inflammatory condition…” , the current sentence is not clear

Line 208: Modify the expression “ RAGE is a sign of infection” because this expression is not clear.

As it was shared with the editor in the first review, could the authors share with the reviewer the images of the other blots that were used for each Western blot, and not just the representative blot. This should be straightforward since the authors repeated all blots several times.

Author Response

We hope that you agree with our manuscript that has been not only thoroughly revised but also strengthened by your comments.

Thank you for your kind consideration.

Line 11: The sentence “RAGE is activated to promote cell death and contributes to … tumors” needs to be modified. In tumors, activation of RAGE results in tumor cell survival, not cell death.

Response: We removed the word “tumor” from the sentence following reviewer comments. Thanks for reviewer’s insightful point.

Line 16: The statement “we investigated the effects of PC on acute renal failure and hepatic dysfunction” is not correct if it refers to the present study. In the current manuscript, the authors only tested the effects of PC on cells. The statement needs to be modified.

Response: We modified the sentence “We investigated the effects of PC on AGEs-induced hepatic and renal cell injury” following reviewer comment.

Line 29-31: The statement “AGEs … are involved in generation infection …” is not correct, please modify.

Response: We modified the sentence “AGEs are produced in the body and interfere with insulin-mediated metabolic reactions and are involved in inflammation-mediated cytokines production through promotion of oxidative stress” following reviewer comment.

Line 49: Please added “the role of” before PC to complete the sentence.

Response: We added “the role of” before PC.

Line 58: Modify the sentence that ends with … “are known to AGEs” In its current form, the sentences is incomplete and doesn’t make sense.

Response: We modified the sentence “RAGE is also involved in various intracellular signaling pathways, and ERK 1/2 and p38 in the mitogen-activated protein kinase (MAPK) pathway, but AGEs are known to be vulnerable to MAPK pathway [16,17].” following reviewer comment. Thanks again for pointing out what we missed.

Line 72: Please confirm that the molecule used to activate RAGE is AGE conjugated with bovine serum albumin, do the authors intended to say that the AGEs were generated from bovine serum albumin? what was the catalog number of this product from BioVision? Please indicate it in the Materials and Methods section. In this same section, please also indicate the catalog number of the anti-RAGE antibody used in Western blots.

Response: We`ve rewritten the confusing sentence and indicated the mentioned catalog numbers (AGEs, RAGE) in the Materials and Methods section. “AGEs (modified with BSA) were purchased from BioVision (2221-10, Milpitas, CA, USA)”, “RAGE antibody was purchased from Santa Cruz Biotechnology (SC-365154, Santa Cruz, CA, USA)”

In general, in all images showing the bands from a WB, please indicate the molecular weight of the band that is shown. Also indicate if this MW is the anticipated MW.

Response: We added molecular weight to all WB band images.

In the legend of Figure 2, c) and d), please replace “inhibition of RAGE activation” by inhibition of RAGE expression.

Response: We changed the sentence following reviewer comment.

Line 180: Please modify the sentence starting with “AGEs-treated causes inflammatory condition…” , the current sentence is not clear

Response: We apologize for the confusion caused by repetition of sentences and incorrect choice of word. We removed the repeated sentence and modified the sentence following reviewer comment.

Line 208: Modify the expression “RAGE is a sign of infection” because this expression is not clear.

Response: We modified the sentence following reviewer comment. “RAGE is a proinflammatory device and transmits inflammatory and risk signals to the body”.

As it was shared with the editor in the first review, could the authors share with the reviewer the images of the other blots that were used for each Western blot, and not just the representative blot. This should be straightforward since the authors repeated all blots several times.

Response: Attached are the results of our another repeated western blots. Thanks again for your outstanding comments for our paper. WB band images are in the PDF file.

Reviewer 2 Report

No further comments

Author Response

Thank you for your kind consideration.

Round 3

Reviewer 1 Report

The manuscript has improved, but there are still requests

Line 29: Please modify the sentence as follows: When AGEs are produced in the body, they interfere with insulin-mediated metabolic reactions.....

Line 56: remove infectious from the expression..." secretion of infectious cytokines"...

Line 60: "vulnerable" is not a suitable word. The sentence is not clear as it is, please modify

Line 185: replace "AGEs-treated" with AGE treatment

Line 212: Replace the sentence by something like..." RAGE activation results in inflammation in the body"....

Author Response

The manuscript has improved, but there are still requests.

We hope that you agree with our manuscript that has been not only thoroughly revised but also strengthened by your comments.

Thank you for your kind consideration.

Line 29: Please modify the sentence as follows: When AGEs are produced in the body, they interfere with insulin-mediated metabolic reactions …

Response: We modified the sentence “When AGEs are produced in the body, they interfere with insulin-mediated metabolic reactions and are involved in inflammation-mediated cytokines production through promotion of oxidative stress [2].” following reviewer comment.

Line 56: remove infections from the expression “secretion of infectious cytokines” …

Response: We removed the word “infections” from the sentence following reviewer comments. Thanks for reviewer’s insightful point.

Line 60: “vulnerable” is not a suitable word. The sentence is not clear as it is, please modify.

Response: We modified the sentence “RAGE is also involved in various intracellular signaling pathways, especially ERK1/2 and p38 are considered to be AGEs-sensitive among the mitogen activated protein kinase (MAPK) pathway [16,17].” following reviewer comment. Thanks again for pointing out what we missed.

Line 185: replace “AGEs-treated” with AGE treatment

Response: We changed the sentence following reviewer comment.

Line 212: Replace the sentence by something like “RAGE activation results in inflammation in the body” …

Response: We modified the sentence following reviewer comment. “As a member of the cell surface pattern-recognition receptors (PRRs), RAGE activation results in inflammation in the body [24].”
